# Socioeconomic Impact Assessment of Water Resources Conservation and Management to Protect Groundwater in Punjab, Pakistan

**Javaria Nasir** [1,*] **, Muhammad Ashfaq** [2]**, Irfan Ahmad Baig** [2]**, Jehangir F. Punthakey** [3,4]**, Richard Culas** [5]**, Asghar Ali** [1,*] **and Faizan ul Hassan** [6]

1   Institute of Agricultural and Resource Economics, University of Agriculture Faisalabad, Faisalabad 38000, Pakistan
2   Department of Agricultural Business and Applied Economics, MNS-University of Agriculture, Multan 61000, Pakistan; ashfaq9@hotmail.com (M.A.); irfan.baig@mnsuam.edu.pk (I.A.B.)
3   Institute for Land Water and Society, Charles Sturt University, Orange, NSW 2800, Australia; eco@ecoseal.com
4   Ecoseal Developments Pty Ltd., Roseville, NSW 2069, Australia
5   School of Agricultural, Environmental and Veterinary Sciences, Institute of Land Water and Society, Charles Sturt University, Orange, NSW 2800, Australia; rculas@csu.edu.au
6   PCRWR, Islamabad 44000, Pakistan; secretary@pcrwr.gov.pk
*   Correspondence: javarianasir@yahoo.com or Javaria.nasir@uaf.edu.pk (J.N.); asghar.ali@uaf.edu.pk (A.A.)

**Abstract:** Water is the most important resource; it is utilized largely in agricultural production and is fundamental to ensuring global food security. This study aims to assess sustainable water management interventions and their impact on the farm economy. To increase water productivity, the most important adaptations that have been proposed are high-efficiency irrigation systems, drought-resistant varieties, the substitution of water-intensive crops with less water-demanding crops, the mulching of soil, zero tillage, and all on-farm operations that can save water, especially ground water. The recent analysis utilized farm survey data from 469 representative farmers along with secondary statistics. The data were collected via a multi-stage sampling technique to ensure the availability of representative farm populations based on a comprehensive site selection criterion. The TOA-MD model estimates the adoption rate of a proposed adaptation based on net farm returns. The impact of high-efficiency irrigation systems and the substitution of high delta crops for low delta crops had a positive impact on net farm returns and per capita income, and a negative impact on farm poverty in the study area. It is recommended that policymakers consult farmer representatives about agricultural and water-related issues so that all the policies can be implemented properly.

**Keywords:** representative agricultural pathways; sustainable agricultural production system; ground water management interventions; farm poverty

## 1. Introduction

Agricultural production systems are complex, interlinked, and play a vital role in global food security. Ground water utilization is an important policy domain in developing nations due to its role in achieving food security and sustainable farming livelihoods. Water is the most important resource that is utilized largely in agricultural production, and the sustainable use of water resources is an important policy objective, as set out in the National Water Policy (2018) and the Punjab Water Policy (2020) [1,2]. Intergenerational and intragenerational equity in terms of farm resources are important to sustainable development [3].

The overutilization of water resources creates complex problems, such as waterlogging and salinization, and results in the depletion of groundwater resources. There are certain planned and unplanned adaptations that can be performed at the farm level and at the farmer's end that can sustain soil fertility and a sustainable farm income. At the

farm/irrigation system level, policy measures, such as the re-allocation of water to higher-value crops and those with limited irrigation requirements, the spatial re-allocation and transfer of water, the adopting of policies that favor the development of water markets, and the efficient utilization of groundwater, can help in improving water productivity in saline environments [4,5].

Agriculture in Pakistan relies on the Indus basin, which is facing severe water scarcity conditions due to climate change. Poor irrigation practices and a lack of policy reforms are major threats for water and food security in the country [6]. This study was designed to analyze the policy impacts at the population level. The research findings could be further utilized in socioeconomic models to analyze their potential effects at the farm level and on farmers' livelihoods [7–9]. Wheat, maize, rice and sugarcane are important food crops in Pakistan, and also play a vital role in food security [10].

Irrigation plays an important role in food and fiber crop production, as 90% of food and 100% of cash crops are directly dependent on irrigation [11]. As an important resource, the exploitation of groundwater and surface water reservoirs must be efficient and sustainable in order to ensure the access to and availability of irrigators [7]. Reforms of policies regarding water and land use are needed in Pakistan for sustainable agricultural production [12]. The federal government of Pakistan aimed to formulate a national water policy in 2018, and suggested different action plans to recover "more crop from each drop". Every drop of water must be utilized in the most efficient manner on-farm, and farmers must utilize efficient irrigation management systems (including crop choice, conveyance and application methods) to ensure aquifer sustainability. Modernization of the irrigation network, participatory management systems, the use of bio-fertilizer and bio-pesticides, efficient water pricing, and salt-tolerant crops must be prioritized for the sustainable use of water resources.

To increase water productivity, the important adaptations proposed include high-efficiency irrigation systems (HEIS), drought-resistant varieties, the substitution of water-intensive crops for less water-demanding crops, zero tillage, and any on-farm operations that can save water, especially ground water [13–17]. This study specifically assesses the policy options, their adoption rates, and their socioeconomic impacts for farming communities, farm production and net farm returns. Two important adaptations were selected for further analysis. These are high-efficiency irrigation systems and the substitution of water-intensive crops with less water-demanding crops.

Representative agricultural pathways (RAPs) are designed to extend global pathways in order to provide the detail needed for regional assessments of future agricultural systems. The Inter-Governmental Panel on Climate Change (IPCC) as developed the concept of the representative concentration pathway (RCP) to develop an understanding of impacts on the environment, water resources and society. Representative agricultural pathways are designed based on a trans-disciplinary process, and are then translated into parameter sets for bio-physical and economic models. RAPs logically precede the definition of management scenarios that embody associated capabilities and challenges. These are designed to be part of a consistent set of drivers and outcomes in national and local developments. Teams of scientists, experts, researchers and farmers with knowledge of agricultural systems work together through a step-wise process [18].

Channelizing the formulation of agricultural policy through the engagement of all possible stakeholders is necessary for the efficacy and better management of natural resources, such as water, soil and air. The formulation of policies in the agricultural sector is interlinked with all sectors of the economy via the forward and backward linkages of the farm economic hierarchy. Research needs to focus on analyzing problems in consultation with farmers and other actors that play an important role in food chains. The practical implementation of researcher's recommendations regarding policy would be increased if specific technical aspects of research findings could be generalized in a simple, non-technical and understandable way, and then communicated to policymakers and other

stakeholders [19]. The links between the agricultural production system and farming livelihood are described in Figure 1.

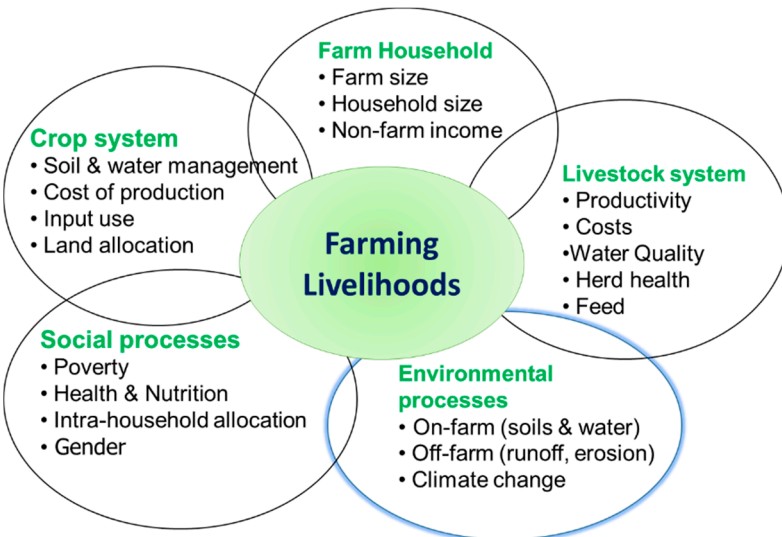

**Figure 1.** The agricultural production system and farming livelihood. Source: [20].

This study is based on a comprehensive engagement process with researchers, policymakers and farmers, investigating the adaptations made at the farm level regarding ground water in order to enhance the sustainable agricultural pathways. The engagement process formulates future agricultural pathways and the economic viability of the proposed adaptations. The interventions impact farm returns, poverty and per capita income under sustainable resource utilization. The adoption of the proposed adaptations and management practices is also evaluated considering economic viability via a cost–benefit analysis. There are very few economic models that can be utilized specifically for policy evaluations. The socioeconomic viability of the recommendations is relevant to the impact of the research and interventions, as the economic rationality influences decisions about the adoption or non-adoption of the proposed adaptations at the farm level [21].

## 2. Materials and Methods

Water is the most critical input required for agricultural production and significantly influences farm ecology. Similarly, bio-physical factors are also important in terms of farm production and livelihoods, via the impact they have on crop choices and production. A recent study was undertaken in the Lower Bari doab[1] (comprised of parts of the Lahore, Sahiwal and Multan divisions) and data from two districts, District Sahiwal and District Okara, have been utilized in the analysis, along with secondary data about the divisions from government statistics. Sahiwal is one of the most fertile divisions in Punjab and is suitable for a variety of cash crops, such as wheat, maize, cotton, rice and sugarcane.

The recent analysis utilized farm survey data collected in 2018 from 469 representative farmers along with secondary statistics for the Sahiwal division. The data were collected by a multi-stage sampling technique to ensure the availability of a representative farm population based on a comprehensive site selection criterion. The distribution of the farm population from the head to the middle and tail ensures the heterogeneous nature of the farms in the data sets. A well-structured questionnaire was developed and personal interviews were conducted to develop a socioeconomic profile of the farmers. RAP sessions were conducted for future projections.

The available data sets were analyzed using the tradeoff analysis multidimensional impact assessment model (TOA-MD), which is a unique simulation tool that can utilize the socioeconomic data sets already collected, combine them with macroeconomic data sets of farms, and project the current and future viability of specific policy interventions proposed

in research studies [22,23]. Due to its efficacy in data use, it is also known as the minimum data approach, as it utilizes secondary data sets for generalizations. The Sahiwal division map is shown in Figure 2.

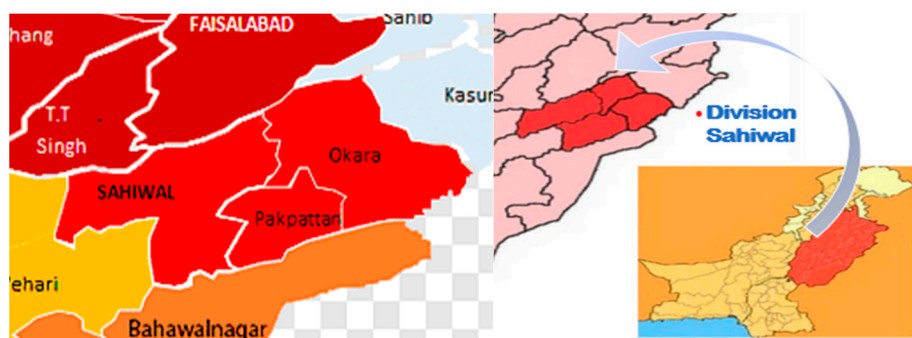

**Figure 2.** Sahiwal division on a map of Pakistan.

*2.1. Representative Agricultural Pathways for Ground Water Management Interventions*

RAPs are formulated via the consideration of representative concentration pathways and shared socioeconomic pathways, as described in Figure 3 below. The shared socioeconomic pathways 1 and 3 are linked with a sustainable development pathway under low and high growth, which are linked with biophysical and socioeconomic indicators. This study develops sustainable development pathways with moderate growth, considering resource depletion and farmer sensitization regarding water conservation and management practices. The shared socioeconomic pathways are determined via the interrelationships of adaptations and conservation practices regarding groundwater in compliance with efforts towards sustainable practices. Shared socioeconomic pathways could be sustainable or unsustainable, with high or low development pathways, and are linked with representative concentration pathways. This study utilized sustainable development pathways and moderate development.

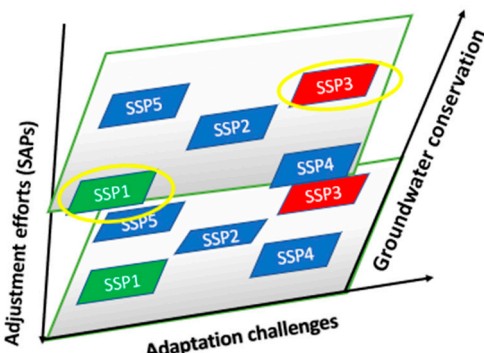

**Figure 3.** Shared socioeconomic pathways and representative concentration pathways. Source: adopted from [24,25].

"Representative agricultural pathways" are combinations of economic, technological and political scenarios that represent a plausible range of possible futures (Box 1). They are not meant to be predictions, but rather provide researchers with a range of plausible scenarios that can be used to simulate possible future outcomes in a consistent and transparent way [26]. The RAPs framework shows that both bio-physical and socioeconomic drivers are essential components of agricultural pathways. RAPs can help engage stakeholders in research throughout the research process, and in the communication and refinement of research results [27].

**Box 1.** The RAP development process. Source: [28].

| |
|---|
| Step 1: Selection of higher-level pathways (Country level) and key indicators identification |
| Step 2: RAPs narratives defined under different shared socioeconomic pathways |
| Step 3: Key parameter/indicator selection and review with consideration of existing literature |
| Step 4: Direction and magnitude of change in variables was shared and comprehensively discussed in RAP meetings |
| Step 5: The rationale for rate of change and a short narrative finalized by a continuous engagement process with experts |
| Step 6: RAPs shared with experts for their feedback |
| Step 7: Feedback from experts and stakeholders in a continuous engagement process incorporated into the refinement of RAPs |
| Step 8: Final RAPs drafted into the RAP matrix and again shared with professionals for further improvement regarding important variables parameterized in the model |

There were three RAP meetings and consultative sessions with stakeholders to formulate the RAPs and water conservation and management practices. The first RAP session was held at PCRWR (Pakistan Council of Research in Water Resources) with hydrological experts, agronomists, social scientists, irrigation scientists and socioeconomic experts. Progressive farmers were also invited to this session. The second consultative session was held in the field, where the experts and the project team visited the farm area and engaged in an extensive group discussion with farmers. The third session was again with academics, and was set up to share and refine the outcome of previous interactive sessions. The RAP parameters' direction and magnitude of change are listed in Table 1.

**Table 1.** Parameterization of RAPs for a sustainable agricultural production system.

| RAPs Key Variables | Direction of Change | Magnitude of Change |
|---|---|---|
| Farm size | Decrease | 10% |
| Household size | Increase | 5% |
| Non-farm income | Increase | 15% |
| Water availability | Increase | 20% |
| HEIS adoption impact on crop yields | Increase | 15% |
| Cost of production | Increase | Impact model projections with RAPs |
| Price | Increase | Impact model projections with RAPs |
| Crop yield | Increase | Impact model projections with RAPs |

Source: Based on consultative session and continuous engagement process.

### 2.2. Propsed Water Conservation and Management Practices

Water conservation and management practices are an important policy domain in terms of food security and sustainable farm livelihoods. Irrigation plays an important role in the production of food and fiber crops. In Pakistan, 90% of food and 100% of cash crops are directly dependent on irrigation [11]. Farmers tend to adopt technologies and conservation techniques as long as they can realize an increase in expected profitability. The adoption of new conservation technologies requires considerable changes in the decision-making process, including human, biophysical, institutional, and economic considerations.

The stakeholder consultative sessions and existing literature suggested these management interventions regarding the sustainable use of water on the farm. Transformative adaptations and the RAP development process are sketched below in Figure 4.

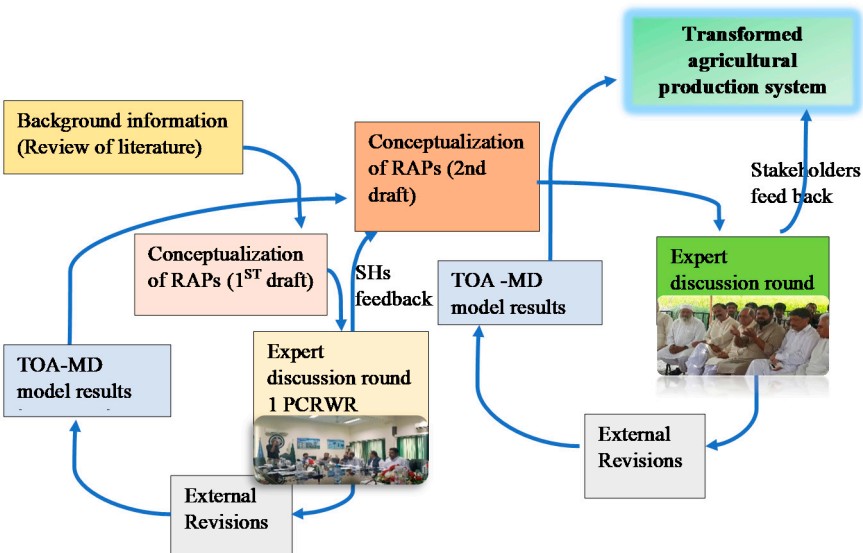

**Figure 4.** Transformative adaptations and RAP development process, which includes three consultative sessions at the Pakistan Council of Research in Water Resources, study area and UAF. Adopted from [29].

By accompanying system-based and problem-solving transformative adaptations, the process adopted in this study, in line with previous studies [30,31], shows that there are complexities in the relations between different actors and contexts of action. Our study pf improved water management and conservation practices suggested the following recommendations [8,32–36]:

- Substitution of high-delta crop with low-delta crop;
- Efficient irrigation practices (HEIS, drip irrigation, sprinklers, etc.);
- Soil conservation practices for better water holding capacity (organic manuring, conservation tillage and cover crops);
- Improved cultivars (drought- and heat-tolerant varieties, short-duration varieties);
- Improved agricultural practices (fertigation, balanced fertilizer, drainage);
- Plant population (seed rate, number of plants);
- Construction of water storage;
- Agricultural insurance/finance;
- Water harvesting.

The above-mentioned strategies are important in terms of water conservation practices and expansions in farm livelihoods. Due to data (availability and quantification) and model limitations, we cannot incorporate all the management interventions at once into the model; therefore, two important interventions proposed and strongly enforced by all stakeholders were used in the model.

The foremost intervention was the substitution of high-delta crops with low-delta crops in the study area. Sustainable farming largely depends on the farm's natural resource capabilities, especially soil fertility and water availability, and quality plays a vital role in terms of farm productivity and income. Most of the farmers are very concerned about the rational use of these resources when selecting the best means of resource exploitation in the long run. Farm tenancy status clearly plays a crucial role in farm management and resource conservation practices. Tenancy status and structure also plays a crucial role in its direct link with sustainable farm livelihoods. However, while farmers are aware of water use and climatic implications, their decisions are largely affected by market signals and public policies. In Pakistan, the government directly intervenes in the wheat market, and provides incentives to wheat farmers in the form of support prices. Wheat is suitable for

most of the irrigated and rainfed areas of Punjab, as it is well suited to the cropping system, and has a stable market with high returns.

Table 2 shows that water consumption under wheat cultivation is the highest, as wheat is preferably grown by farmers due to its better returns, proper marketing, efficient value chain and seed availability, but it utilizes the most water—a reported 39 million acre-feet [37].

**Table 2.** Current water consumption by five major crops.

| Crop Water Consumption | Million Acre Feet (MAF) |
| --- | --- |
| Wheat | 39 |
| Cotton | 29 |
| Rice | 26 |
| Sugarcane | 23 |
| Maize | 5 |

Source: [25].

The research regarding ground water management interventions has been described in Figure 5.

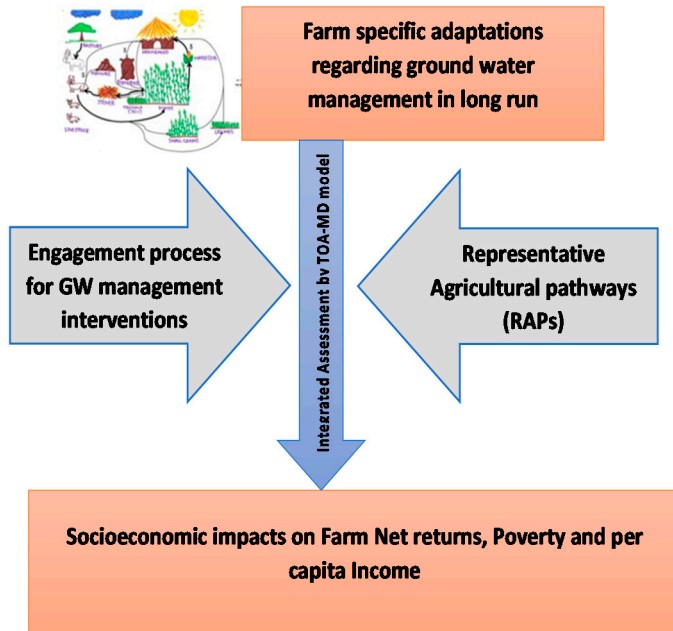

**Figure 5.** Research framework regarding ground water management interventions.

*2.3. Tradeoff Analysis Multidimensional Impact Assessment Model*

The TOA-MD simulates various "experiments" for the adaptation of new technologies and their impact assessments. These "experiments", combined with scenarios that represent the state of the world (for example, current or future technology), are the basis of adaptation analysis [38].

Current = Current technology
Future = Future (changed) technology
Adapted = Adapted (changed) technology
Consider the two scenario systems 1 and 2:
system 1 = Current time period, base technology
system 2 = Future time period, improved technology (proposed interventions)
w = v1 − v2 = Measures the difference in income with base and changed technology

if w > 0 Technology adoption causes a loss

w < 0 Technology adoption causes a gain

So, we can interpret the adoption model as:

Adopters = Those who gain from technology adoption (a farmer who would like to "adopt" new technology).

Non-adopters = Those who suffer from technology adoption (farmers who would not like to "adopt" a new technology).

The "adoption rate" at a = 0 separates losers from gainers.

Figure 6 describes the possible development pathways and adaptation options in the future. This study adopts the sustainable development pathways as projected by the green pathway, and the increases in farm incomes resulting from the adoption of management interventions are shown. The adoption rate for proposed interventions is estimated via the opportunity cost of adoption and non-adoption. The difference between net returns from system 1 and system 2 defines the opportunity cost.

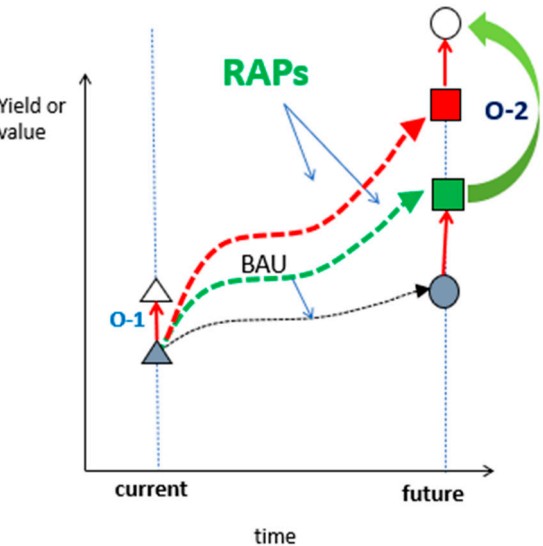

**Figure 6.** Pathways and future production systems. Source: [6,10].

$$\omega = \upsilon1 - \upsilon2 \tag{1}$$

where $\upsilon1$ = net returns from system 1; $\upsilon2$ = net returns from system 2.

$\omega < 0$ means gain from adoption of system 2

$\omega > 0$ means loss from adoption of system 2

Thus, the proportion of adopters is given by:

$$A = 100 \int_{-\infty}^{0} \varphi(\omega) d\omega \tag{2}$$

The percentage of non-adopters or farmers that remain in system 1 is:

$$\Lambda = 100 - 100 \int_{-\infty}^{0} \varphi(\omega) d\omega = 100 \int_{0}^{\infty} \varphi(\omega) d\omega \tag{3}$$

## 3. Results

The results of the TOA-MD model project the adoption rate based on the economic viability of the proposed adaptation strategies in the study area. The important aspect of this analysis is the formulation of an adaptation package, and the further selection of those management interventions that could be analyzed for economic viability. The TOA-MD model is used to analyze the adoption rate of the proposed management interventions

based on economic analysis and cost–benefit analysis. The input for the model is based on survey data and projections of RAPs. The output of the model is mainly the adoption rate of adaptations based on impact on per capita income, net farm returns and farm poverty.

The TOA-MD model was used to estimate the adoption rate of the proposed adaptation based on farm returns, and the impact of the substitution of high-delta crops with low-delta crops on net farm returns, per capita income and poverty in the study area.

*Economic Benefits for Adoption of Low Delta Crops*

The proposed adaptation package includes the substitution of high-delta crops with low-delta crops. According to the cropping pattern of the study area, the proposed crops are oilseed and pulses, which may increase the water productivity and farm livelihood, and reduce the import burden on the economy. The suggested management interventions include crop diversification; 5–10% of the area under wheat crop is replaced by sunflower, and likewise, 5–10% of the maize could be replaced with moong bean. Sunflower and moong bean are highly recommended crops according to the climatic and biophysical condition of the study area. Additional data sets were collected for alternative crops in the same area in order to analyze the socioeconomic impacts of crop substitution. These could be added to oilseed and pulses in the analysis to check their economic viability. The rationale of crop substitution is based on resource conservation and export bills; likewise, crop diversification also protects farmers from risk and uncertainty, and the adoption of legumes crops can improve soil fertility and crop productivity, as well as using less water. There are certain risks involved, especially market risks and climatic uncertainties [39,40]. The proposed interventions include the repurposing of 5–10% of the land for alternate crops that increase farm returns and ensure increased farm incomes. Wheat is a staple crop that utilizes very high water resources, but this crop has huge socioeconomic and political importance. The consultative session suggested that annual surplus wheat production could be replaced by oilseed and pulses [40]. However, the farm sizes are already small, so it was agreed that 5% of land occupied by small land holdings and 10% of land occupied by large land holdings could be allocated for alternate crops. The economic benefits of the adoption of low-delta crops are described in Table 3.

**Table 3.** Economic benefits of the adoption of low-delta crops.

| Scenarios | Crop Substitution | Percentage Change in Net Farm Returns | Percentage Change in per Capita Income | Change in Poverty | Percentage Potential Adoption Rate |
|---|---|---|---|---|---|
| Scenarios 1 | Maize replaced by moong bean | 32.51 | 28.75 | −3.6 | 49 |
| Scenarios 2 | Wheat replaced by sunflower | 33.89 | 30.05 | −3.4 | 59 |

Note: Poverty line was calculated based on USD 1/person/season, and was INR 21900 for one season.

The results indicate that, without adaptation, the poverty level was 17.72% in the survey data, whereas with interventions, the poverty rate would reduce by 3.6 and 3.4% for scenario 1 and 2, respectively. The percentage adoption rates for sunflower and moong bean would be 49 and 59%, respectively, as described in Figure 7. The results show that the proposed interventions, such as crop substitution, would have a significant impact on farm livelihoods.

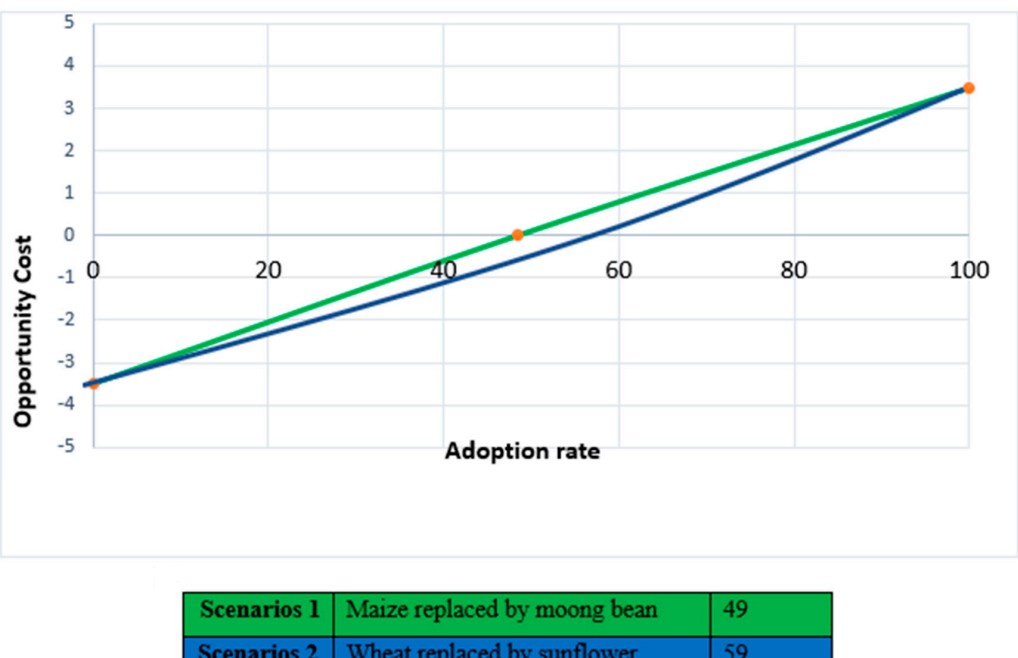

| Scenarios 1 | Maize replaced by moong bean | 49 |
|---|---|---|
| Scenarios 2 | Wheat replaced by sunflower | 59 |

**Figure 7.** Potential adoption rate of low-delta crops in study area for a future agricultural production system.

The percentage change in net farm returns per capita income and farm poverty due to crop substitution is shown in Figure 8.

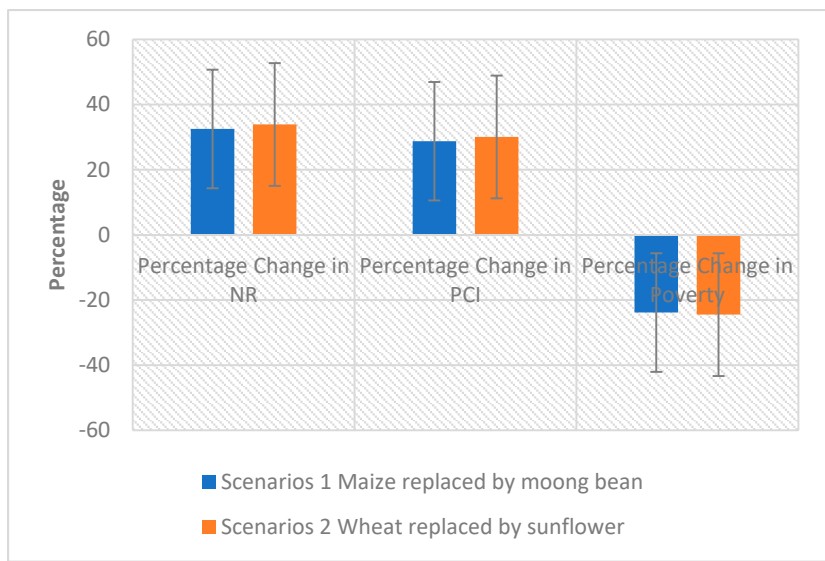

**Figure 8.** Percentage change in net returns, per capita income and poverty due to crop substitution.

Farmers tend to adopt technologies and conservation techniques as long as they can realize an increase in expected profitability. The decision to adopt technologies and techniques is also influenced by a farmer's socioeconomic status, knowledge of new technologies, cultural background, and access to natural resources. Moreover, the adoption of new conservation technologies requires considerable changes in the decision-making process, including a wide range of human, biophysical, institutional, and economic considerations.

The socioeconomic impacts of the benefits offered by a high-efficiency irrigation system are presented in Table 4.

**Table 4.** Socioeconomic impacts of benefits offered by a high-efficiency irrigation system.

| Crops | % Change in Net Returns | % Change in per Capita Income | Change in Poverty | % Potential Adoption Rate |
|---|---|---|---|---|
| Cotton | 28.307 | 23.002 | −4.7 | 72.000 |
| Maize | 40.804 | 36.586 | −4.6 | 76.026 |
| Wheat | 19.821 | 18.052 | −1.0 | 63.128 |
| Rice | 37.947 | 31.410 | −5.3 | 87.036 |
| Sugarcane | 37.943 | 35.362 | −3.6 | 55.775 |
| Overall | 34.893 | 30.564 | −4.7 | 73.800 |

Note: Calculations are based on RAPs and TOA-MD calculations.

The proposed interventions include the substitution of conventional methods of irrigation with high-efficiency irrigation systems (HEISs). With the proposed intervention of HEISs (through technology and management), there would be 3.8% decrease in poverty and a 30% increase in per capita income. The adoption rate of this adaptation package is 74%, resulting in a reduction in farm poverty, as presented in Figure 9 It is evident that substantial reductions in water consumption are made possible through changes in cropping patterns, and the suggested crops increase the farm income and livelihood as well. Low-delta crops must be prioritized as compared to high-delta crops, considering the demand for staple foods to ensure food security. The adoption rate of high-efficiency irrigation systems for major cash crops has a positive impact on NR, PCI and farm poverty, as shown in Figure 9.

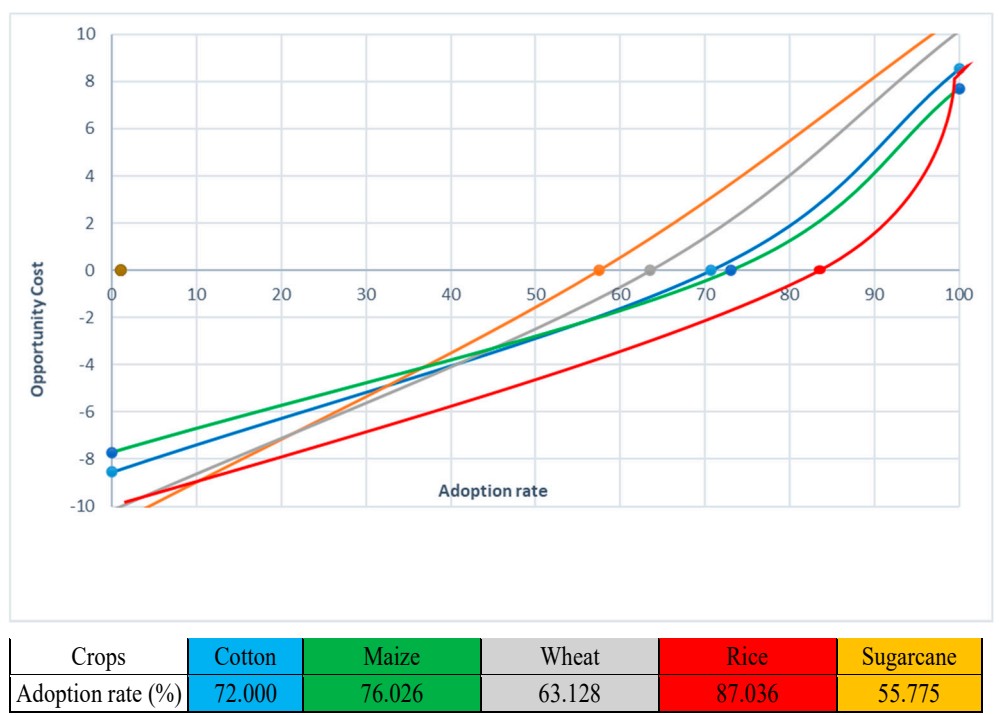

| Crops | Cotton | Maize | Wheat | Rice | Sugarcane |
|---|---|---|---|---|---|
| Adoption rate (%) | 72.000 | 76.026 | 63.128 | 87.036 | 55.775 |

**Figure 9.** Adoption rate of high-efficiency irrigation systems for major cash crops.

The result further shows that water savings and high NR are made possible by shifting from conventional irrigation to improved irrigation technologies (sprinkler and drip irrigation) [34]. Public policies must consider resource conservation and sustainable livelihoods. Policies must be formulated in favor of institutional development, as compared to support for domestic production and price control policies [4]. Based on the following results, it is recommended that apart from the proposed water-saving strategies, other

alternative management techniques, directed to off-farm (i.e., improved infrastructure to reduce water losses due to poor conveyance efficiency) and on-farm (e.g., deficient irrigation or soil mulching) management, should be evaluated in future studies, as these play important roles in the sustainable use of farm resources and farm livelihoods. Figure 10 describe the impact of the adoption of HEIS on net farm returns, per capita income is positive but negative impact on farm poverty.

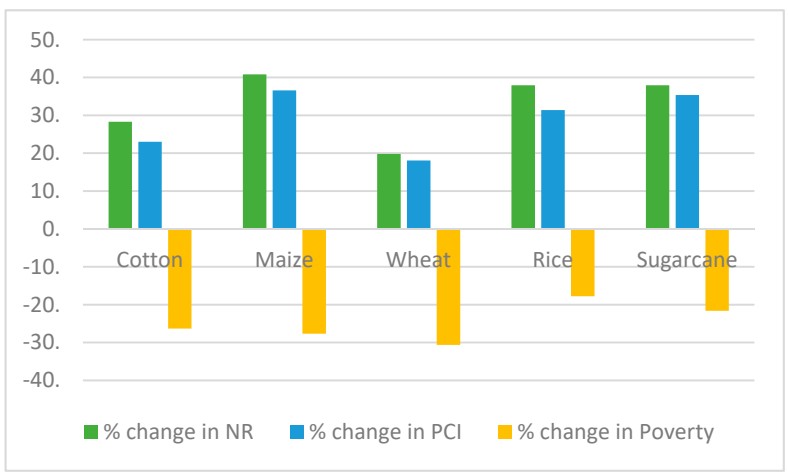

**Figure 10.** Impact of the adoption of HEIS on net farm returns, per capita income and poverty.

## 4. Discussion

Agricultural production systems could be improved by educating farmers to adopt management interventions and improved practices that are practically possible at the farm level. The main concern with the proposed interventions and solutions at research stations is the slow adoption and the economic viability of the farm. As such, the adaptations are at times not practically adopted by farmers, due to technological, financial, and socioeconomic constraints [41–44]. As farmers are the most important and crucial stakeholders in the whole process, it is important to involve the farmer in the entire research process when devising solutions for the agricultural issues that farmers face in the long run [45,46].

To acknowledge the importance of agricultural production systems in society, studies describe the effects of transformative adaptations on boosting farm produce [28,37,38]. This project evaluates the potential adaptations that can be made in the study area, and also emphasizes the reasons for low adoption due to the lack of important stakeholders involved in the whole policy formation process. The researchers, policymakers and farmers working in their specialized fields maintain weak communication. During the RAP sessions, researchers from sociology, genetics, irrigation and drainage, economics and the soil sciences were consulted in developing improved projections.

From a food security perspective, Pakistan's agricultural policy mainly concentrates on wheat, especially in terms of support price and procurement. This ensures farm returns to some extent, but creates inefficiencies in the wheat market. Wheat is a suitable crop all over Pakistan, and fits better in all cropping systems. However, wheat crops compete with the main oilseed crops and pulses. Research studies have shown that sunflower is suitable for cultivation and is a highly profitable crop; however, farmers are reluctant given several constraints [47,48]. The adoption of sunflower is low, mainly due to seed unavailability, the high cost of production (especially seed cost), inefficient marketing, the lack of suitable farm machinery for small farms, and the lack of competition among buyers [49].

The allocation of water resources is crucial in terms of sustainable agriculture; consequently, policies must be formulated for water conservation and management. The formulation and implementation of ground and surface water laws are linked with the adoption of water conservation practices, such as the implementation of micro-irrigation technologies and the growing of high-value crops, which can boost the water economy [50].

Pakistan's farming landscapes are complex and varied across regions in terms of water availability, quality and quantity. Therefore, it is recommended to provide water-related information that is authentic and constructed with reference to all the specific policies devised and implemented to enable sustainability in farming practices [9]. The on-farm management practices, such as the use of fertilizer, soil management, laser leveling, irrigation methods and the selection of crops, contribute to water use efficiency, soil health improvement and the mitigation of climate change impacts [7,30,31].

The moong bean has great potential as a cash crop in Pakistan in wheat systems, but there are certain interventions that can turn constraints into opportunities in terms of its adoption in irrigated areas [51]. The availability of water, high-yielding cultivars, improved management practices and improvements in the value chain are the crucial factors in moong bean cultivation. The cultivation of pulses, especially moong bean, is low, mainly due to marketing issues and inconsistent policies. The benefit–cost ratio of moong bean is higher than that of all other major cash crops in certain areas, as reported by the National Agricultural Research Centre, but due to marketing factors, farmers do not grow this crop, and are reluctant to substitute existing crops [48].

The improvement in yield is substantial when adopting efficient irrigation systems, especially in maize crop [52]. The biological and grain yields increased substantially with higher-efficiency irrigation systems and raised bed systems. The water quality and mode of irrigation could increase the crop yields by approximately 15%. Improvements in water quality and soil fertility increase the crop yields substantially. It is projected that better on-farm water management increases the net farm income substantially by increasing the crop yields and reducing the cost of production—it is estimated that this increases the farm income by INR 75000 per acre per annum [43,53].

## 5. Conclusions

Agricultural production systems largely depend on natural resources, especially water and soil. On-farm water conservation and management practices are needed, with measures such as the re-allocation of water to higher-value crops. Likewise, limited irrigation requirements, spatial re-allocation and the transfer of water improve water productivity, and have positive impacts on farm livelihoods. The adaptations that could increase water productivity formulated during the engagement process include high-efficiency irrigation systems (HEIS), drought-resistant varieties, the substitution of water-intensive crops with less water-demanding crops, the mulching of soil, zero tillage, and improved farm cultivation operations.

Overall, 75% of farmers have the economic ability to adopt these management interventions. Although wheat, maize, rice, sugarcane and cotton are the most important cash crops, it is necessary to calculate the social cost of water-demanding crops. Oilseeds and pulses are potential candidates in terms of resource conservation and crop diversification.

Based on current analysis, the following recommendations could be helpful for researchers and policymakers to improve water management. Policy formulation must consider and consult farmer representatives about water issues, so that all the policies can be implemented at the farm. Farmers are important stakeholders, and their inclusion will help in the adoption of interventions to improve the management of ground water and surface water.

The most crucial factor in agricultural development is the access to agricultural finance, especially for the adaptation of technological advancements regarding water management, such sprinkler and drip irrigation and farm mechanization. The study recommends that the Central Bank provide special financing schemes for sustainable practices, which would increase the rate of adoption.

Water allocation at the farm must be equitable, and there must be an efficient water market so that malpractices and the overutilization of water resources can be minimized. Nature-based solutions are also needed, and appropriate policies must be formulated for

specific zones. The areas with water scarcity must be highlighted, and serious efforts should be made to implement the suggested interventions.

The integrated farm system model is a concept proposed to increase the farming systems' efficiency in a sustainable manner. Livestock, crops, fisheries, agroforestry and the poultry sector must all be considered as one integrated system. Public policies that are in favor of one crop could suppress the cultivation of other crops. Support prices for a farming system could be avoided, as themarket-based solutions are the most efficient. Public policies that are in favor of one crop could suppress the cultivation of other crops. Therefore, public support especially the support prices should be designed and implemented in a way to enhance welfare gain for farming system as a whole whereas efficient market-based solutions should be preferred. For the up-scaling of the interventions, it is recommended that the current work be continued in other agro-ecological zones of Punjab where the majority of farmers and non-farmers are resource-poor, water is scarce, and poverty is high. Solving issues in the community by involving its members is very important, and RAPs are a novel approach to providing solutions for critical issues and later assessing the impacts on livelihood of the implemented interventions.

The analysis was conducted for the mid-century; the future is unpredictable, and many development pathways and future parameters of sustainability could change the extent of the impacts. This study only considers the sustainable development pathways, and assumes that sufficient effort would be made toward resource conservation. There is a lack of data on the region- and crop-specific ex ante analytical impact of water quality and quantity. Alternate livestock and horticultural crops were not analyzed due to the lack of survey data, although it this be a potential future enterprise in the study area.

This analysis could be further refined by considering adaptations of water harvesting, storage, and water prices, which are important indicators related to the efficient use of water and agricultural resources for other areas of Pakistan. The analysis could be performed using more than one pathway and future price assumptions.

**Author Contributions:** Conceptualization, J.N. and M.A.; methodology, J.N. and I.A.B.; software, J.N.; validation, M.A., J.N. and J.F.P.; formal analysis, J.N.; investigation, J.N.; resources, J.F.P.; data curation, A.A.; writing—original draft preparation, J.N.; writing—review and editing, J.N., R.C., A.A., F.H.; visualization, J.N.; supervision, M.A.; project administration, J.F.P., A.A., R.C.; funding acquisition, J.F.P., F.u.H. All authors have read and agreed to the published version of the manuscript.

**Funding:** This research was funded by the Australian Centre for International Agricultural Research (ACIAR), grant number LWR-2015/036.

**Institutional Review Board Statement:** Not Applicable.

**Informed Consent Statement:** Informed consent was obtained from all subjects involved in the study.

**Data Availability Statement:** The data will be available upon request.

**Acknowledgments:** This research was largely contributed by ACIAR project team and all team stakeholders including the farming community that was on board throughout the research process.

**Conflicts of Interest:** The authors declare no conflict of interest.

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
