# Peer review of "Socioeconomic Impact Assessment of Water Resources Conservation and Management to Protect Groundwater in Punjab, Pakistan"

_water, doi:10.3390/w13192672_

Round 1
Reviewer 1 Report
The study assesses the socioeconomic impact of ground water management interventions through representative agricultural pathways for mid-century in Punjab-Pakistan using TOA-MD 19 model. I found the paper interesting and the authors presented their findings very well. I believe the paper will be a good addition to Water journal. However, the manuscript requires some moderate modifications before publication. Please find more details in the second comment below.
Major Issues:
- Line 86- Figure1: Please explain why the (SSP1 and SSP3) are circled in the figure.
- The authors should summarise the main goals of their study in a short paragraph by the end of the introduction section.
- Please explain more about “secondary data sets used in this study”. What the authors mean exactly about this data sets?
- Please draw a flow chart for the TOA-MD model procedure in the methodology section to understand clearly how this tool works. I believe it is very important to show the inputs and outputs of the model like how it was explained in the results section: “The input of model is based on survey data and projections of RAPs. 248 The output of model is mainly adoption rate of adaptations based on impact on per capita 249 income, net farm returns and farm Poverty.”.
- Please double check your references as I couldn’t find the following references in the manuscript: [16,25,40,44, and 46].
Minor Issues:
- Line 278: Please define NR and PCI where they first occur in the manuscript (Table 3).
- Line 241: Please add reference for Table 2.
- Line 131: Please define TOA-MD where it first occurs in the manuscript.
- Line239-240: Please check the alignment of the lines.
- Line 242: Please delete the unneeded parenthesis in the title.

Author Response
Thanks for the valuable input, we added all possible comments suggestions in the paper for your kind consideration.

Reviewer 2 Report
The authors present an overview of the case for changing cropping patterns in the Punjab based upon an analysis using the Trade-off Analysis Model for Multi-dimensional Impact Assessment (TOA-MD). The manuscript is very uneven with upper case letters seemingly randomly inserted--there is little consistency in the use of upper case letters for terms used. TOA-MD is not defined, and the referencing has simply replaced the Harvard system with numbers so that the referencing is not sequential. In places, the Harvard system is still used: line 52 should cite [48] (using the current reference list) and line 350 should cite [34]. Citations are incomplete in the list, using an ellipse instead of the full authorship list (as in citations [5], [6], [33], [40], [45], and [47]. The author affiliations are incomplete (lines 6-10). The entire manuscript should be reviewed for standard English usage (especially the use of articles--a, an, and the).
The TOA-MD model is not well described, with the text on lines 131-132 duplicating lines 139-140 without actually describing the modelling process. Are focus groups used? If so, how were they formed; how many times did they meet; and, how were decisions reached that were included in the model? On line 159, a closing parenthesis is needed.
On lines 178-180, punctuation is needed to clearly show the three parts, given that each part has multiple components. On line 207 the ampersand should be replaced by "and", and on line 210 "no" should be restated as "number". On line 218, should there be a "not" after the word "can"?
Lines 251-253 seem to be incomplete (or the punctuation needs to be reviewed). There also seems to be missing text on lines 264 and 361.
The terms NR and PCI in Tables 3 and 3.9 need to be defined--Table 3.9 should be Table 4? Figures 5.3 and 5.4 should be Figures 8 and 9? In Figure 5.4, there are too many zeros associated with the Y axis.
The spelling of "moong" and "mung" beans needs to be consistent. Water quality on line 389 needs to be defined--presumably it related to salinity? The mention of "nature-based solutions" on line 424 seems to be the only mention of this concept? Further, where there are mentions of topics such as a "famous livestock breed" (line 125) and "certain constraints" (lines 344 and 361) examples should be mentioned as readers may not be familiar with the region under discussion.
In short, the manuscript needs to be thoroughly reviewed and revised prior to further consideration: portions seem to have been abstracted from a report while other parts reflect a very uneven writing style.
Author Response
Please find the attached comments that was added according to your guidelines.
Best Regards,

Reviewer 3 Report
Dear Authors
Here I send you proposals for manuscript improvement.
Title
Change the title.
- mid-century (of which century?)
- groundwater management (I would say agriculture land management to protect groundwater resources
From: Socioeconomic impact assessment of ground water management interventions through representative agricultural pathways for mid-century in Punjab-Pakistan
To: Socioeconomic impact assessment of water resources conservation and management to protect groundwater in Punjab, Pakistan
Abstract:
Add in the abstract aim or objective of the study.
line 16: "mulching soil" - that is not sustainable - replace.
1. Introduction:
line 66: "mulching soil" - that is not sustainable - replace.
line 115: add the aim or objective of the study.
2. Materials and Methods
Extend this chapter and structure this chapter.
2.1 Case study description (117-125)
extend the text
2.2 Database and survey design (126-136)
extend this chapter, explain more in detail how data was collected. if there was a questionnaire, workshops, etc.
2.3 Representative agricultural Pathways
-move here text between lines 72-115.
2.4 TOA-MD model (line 139-175)
2.5 Uncertainties and shortcomings
explain what could be done better if more data would be. were are weak points of study.
3. Results
3.1 - line 181
half of this text should be in the materials.
Description of methods and materials has to be moved chapter 2.
Figure 5 is something that would be needed in subchapter 2.2.
In this chapter, you need to present the results of this process.
Figure 5 all abbreviations need to be explained in the figure title. All figures and tables must be self-explanatory also when it stands alone.
Present here which practices were defined as important by farmers. Present some statistics. Which age group or farm size is keener to implement the specific practice.
3.2 - line 239
I am missing text here. You need to explain the table.
3.3 - line 242
do not repeat the text that should be in Materials and methods (line 243-253).
Repurpose or delete this text.
3.3.1 - line 254.
These are the real results.
Table 3: Explain abbreviations in the table. Add units.
Rupees should be replaced by USD, mark the conversion rate and date.
Figure 6: you mention here scenarios.
If there were scenarios the need to be explained in 2. Materials and method (explain the scenarios). The readers do not know what the mid-century scenario means.
Renumber figures and tables with single digit.
line 293 - is this figure or table
line 306 - table or title of chapter - explain abbreviations.
5. Conclusions
Currently is here only repeated text.
Replace with answers to these questions:
Why is this research unique?
What are the shortcomings/uncertainties of this research?
What did the scientific community learn out of it?
What are the benefits/recommendations for stakeholders (farmers, water managers)?
What are the recommendations for policymakers/legislators?
Future work?
Author Response
Please find the response towards your review on paper, I try to address all possible suggestions/ comments for your kind consideration.
Regards,

Round 2
Reviewer 2 Report
The authors have substantially revised their manuscript. Nevertheless, it is recommended that they take one more pass through to address a few outstanding concerns.
On lines 34 and 35: the referenced water laws should be listed in the references and an appropriate numeric reference assigned.
On line 54, the Harvard reference to Yang et al should be referenced as [48].
On line 95, the full stop should be transposed with the figure number.
Footnote 1: this footnote is replicated throughout the manuscript; once is enough.
Figure 3 needs further explanation; it is not clear from the caption or the text what is intended to be illustrated.
Line 141: pathways 1 and 3 need further explanation as there are no clear pathways shown.
Line 164: define PCRWR here even though it seems to have been defined in the caption to Figure 4, line 189.
Table 1, column 2, row 5: delete the "d" for consistency.
Line 221: this seems to be the first mention of "farm tenancy status" so further explantion would seem appropriate. It makes sense that this is a factor since ownership would provide the farmer with more options.
Figure 6: This figure should be discussed/explained and the acronyms and abbreviations defined in the caption. As it stands, it lacks context and relevance.
Line 275: delete the "e" from project.
Table 3: use "Scenario" on rows 2 and 3 in column 1. The Note to the figure (line 310) is missing text at the end of the line: "(1$-" ???.
Line 346: the percentages are derived from discussions with farmers and others, so perhaps it would be better to state the percentages as "estimated"?
Lines 471-472: the sentence is incomplete.
Please also review the manuscript to ensure number agreement between the noun and verb forms; frequently, singular nouns have plural verb forms.
Author Response
Dear,
We try to incorporate all your suggestions and improvements.
Best regards,
Dr. Javaria
